# Meta-Learning with Implicit Gradients

**Aravind Rajeswaran**∗
University of Washington
aravraj@cs.washington.edu

**Chelsea Finn**∗
University of California Berkeley
cbfinn@cs.stanford.edu

**Sham M. Kakade**
University of Washington
sham@cs.washington.edu

**Sergey Levine**
University of California Berkeley
svlevine@eecs.berkeley.edu

## Abstract

A core capability of intelligent systems is the ability to quickly learn new tasks by drawing on prior experience. Gradient (or optimization) based meta-learning has recently emerged as an effective approach for few-shot learning. In this formulation, meta-parameters are learned in the outer loop, while task-specific models are learned in the inner-loop, by using only a small amount of data from the current task. A key challenge in scaling these approaches is the need to differentiate through the inner loop learning process, which can impose considerable computational and memory burdens. By drawing upon implicit differentiation, we develop the implicit MAML algorithm, which depends only on the solution to the inner level optimization and not the path taken by the inner loop optimizer. This effectively decouples the meta-gradient computation from the choice of inner loop optimizer. As a result, our approach is agnostic to the choice of inner loop optimizer and can gracefully handle many gradient steps without vanishing gradients or memory constraints. Theoretically, we prove that implicit MAML can compute accurate meta-gradients with a memory footprint no more than that which is required to compute a single inner loop gradient and at no overall increase in the total computational cost. Experimentally, we show that these benefits of implicit MAML translate into empirical gains on few-shot image recognition benchmarks.

## 1 Introduction

A core aspect of intelligence is the ability to quickly learn new tasks by drawing upon prior experience from related tasks. Recent work has studied how meta-learning algorithms [51, 55, 41] can acquire such a capability by learning to efficiently learn a range of tasks, thereby enabling learning of a new task with as little as a single example [50, 57, 15]. Meta-learning algorithms can be framed in terms of recurrent [25, 50, 48] or attention-based [57, 38] models that are trained via a meta-learning objective, to essentially encapsulate the learned learning procedure in the parameters of a neural network. An alternative formulation is to frame meta-learning as a bi-level optimization procedure [35, 15], where the "inner" optimization represents adaptation to a given task, and the "outer" objective is the meta-training objective. Such a formulation can be used to learn the initial parameters of a model such that optimizing from this initialization leads to fast adaptation and generalization. In this work, we focus on this class of optimization-based methods, and in particular the model-agnostic meta-learning (MAML) formulation [15]. MAML has been shown to be as expressive as black-box approaches [14], is applicable to a broad range of settings [16, 37, 1, 18], and recovers a convergent and consistent optimization procedure [13].

---

∗Equal Contributions. Project page: http://sites.google.com/view/imaml

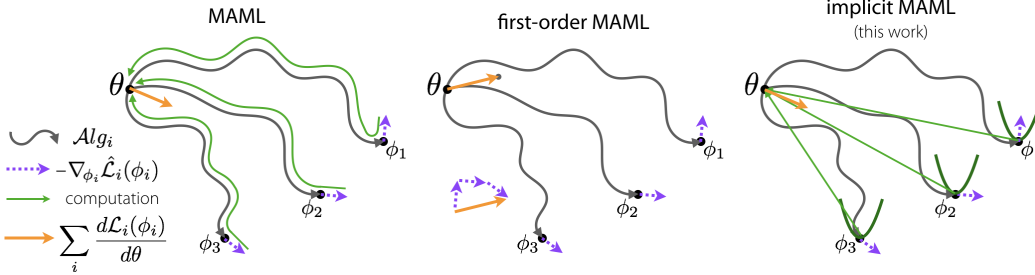

Figure 1: To compute the meta-gradient $\sum_i \frac{d\mathcal{L}_i(\phi_i)}{d\theta}$, the MAML algorithm differentiates through the optimization path, as shown in green, while first-order MAML computes the meta-gradient by approximating $\frac{d\phi_i}{d\theta}$ as $\boldsymbol{I}$. Our implicit MAML approach derives an analytic expression for the exact meta-gradient without differentiating through the optimization path by estimating local curvature.

Despite its appealing properties, meta-learning an initialization requires backpropagation through the inner optimization process. As a result, the meta-learning process requires higher-order derivatives, imposes a non-trivial computational and memory burden, and can suffer from vanishing gradients. These limitations make it harder to scale optimization-based meta learning methods to tasks involving medium or large datasets, or those that require many inner-loop optimization steps. Our goal is to develop an algorithm that addresses these limitations.

The main contribution of our work is the development of the implicit MAML (iMAML) algorithm, an approach for optimization-based meta-learning with deep neural networks that removes the need for differentiating through the optimization path. Our algorithm aims to learn a set of parameters such that an optimization algorithm that is initialized at and regularized to this parameter vector leads to good generalization for a variety of learning tasks. By leveraging the implicit differentiation approach, we derive an analytical expression for the meta (or outer level) gradient that depends only on the solution to the inner optimization and not the path taken by the inner optimization algorithm, as depicted in Figure 1. This decoupling of meta-gradient computation and choice of inner level optimizer has a number of appealing properties.

First, the inner optimization path need not be stored nor differentiated through, thereby making implicit MAML memory efficient and scalable to a large number of inner optimization steps. Second, implicit MAML is agnostic to the inner optimization method used, as long as it can find an approximate solution to the inner-level optimization problem. This permits the use of higher-order methods, and in principle even non-differentiable optimization methods or components like sample-based optimization, line-search, or those provided by proprietary software (e.g. Gurobi). Finally, we also provide the first (to our knowledge) non-asymptotic theoretical analysis of bi-level optimization. We show that an $\epsilon$–approximate meta-gradient can be computed via implicit MAML using $\tilde{O}(\log(1/\epsilon))$ gradient evaluations and $\tilde{O}(1)$ memory, meaning the memory required does not grow with number of gradient steps.

## 2 Problem Formulation and Notations

We first present the meta-learning problem in the context of few-shot supervised learning, and then generalize the notation to aid the rest of the exposition in the paper.

### 2.1 Review of Few-Shot Supervised Learning and MAML

In this setting, we have a collection of meta-training tasks $\{\mathcal{T}_i\}_{i=1}^M$ drawn from $P(\mathcal{T})$. Each task $\mathcal{T}_i$ is associated with a dataset $\mathcal{D}_i$, from which we can sample two disjoint sets: $\mathcal{D}_i^{\mathrm{tr}}$ and $\mathcal{D}_i^{\mathrm{test}}$. These datasets each consist of $K$ input-output pairs. Let $\mathbf{x} \in \mathcal{X}$ and $\mathbf{y} \in \mathcal{Y}$ denote inputs and outputs, respectively. The datasets take the form $\mathcal{D}_i^{\mathrm{tr}} = \{(\mathbf{x}_i^k, \mathbf{y}_i^k)\}_{k=1}^K$, and similarly for $\mathcal{D}_i^{\mathrm{test}}$. We are interested in learning models of the form $h_\phi(\mathbf{x}) : \mathcal{X} \rightarrow \mathcal{Y}$, parameterized by $\phi \in \Phi \equiv \mathbb{R}^d$. Performance on a task is specified by a loss function, such as the cross entropy or squared error loss. We will write the loss function in the form $\mathcal{L}(\phi, \mathcal{D})$, as a function of a parameter vector and dataset. The goal for task $\mathcal{T}_i$ is to learn task-specific parameters $\phi_i$ using $\mathcal{D}_i^{\mathrm{tr}}$ such that we can minimize the population or test loss of the task, $\mathcal{L}(\phi_i, \mathcal{D}_i^{\mathrm{test}})$.

In the general bi-level meta-learning setup, we consider a space of algorithms that compute task-specific parameters using a set of meta-parameters $\boldsymbol{\theta} \in \Theta \equiv \mathbb{R}^d$ and the training dataset from the task, such that $\boldsymbol{\phi}_i = \mathcal{A}lg(\boldsymbol{\theta}, \mathcal{D}_i^{\mathrm{tr}})$ for task $\mathcal{T}_i$. The goal of meta-learning is to learn meta-parameters that produce good task specific parameters after adaptation, as specified below:

$$\overbrace{\boldsymbol{\theta}_{\mathrm{ML}}^* := \operatorname*{argmin}_{\boldsymbol{\theta} \in \Theta} F(\boldsymbol{\theta})}^{\mathrm{outer-level}}, \text{ where } F(\boldsymbol{\theta}) = \frac{1}{M} \sum_{i=1}^{M} \mathcal{L}\bigg( \overbrace{\mathcal{A}lg(\boldsymbol{\theta}, \mathcal{D}_i^{\mathrm{tr}})}^{\mathrm{inner-level}}, \mathcal{D}_i^{\mathrm{test}} \bigg). \tag{1}$$

We view this as a bi-level optimization problem since we typically interpret $\mathcal{A}lg(\boldsymbol{\theta}, \mathcal{D}_i^{\mathrm{tr}})$ as either explicitly or implicitly solving an underlying optimization problem. At meta-test (deployment) time, when presented with a dataset $\mathcal{D}_j^{\mathrm{tr}}$ corresponding to a new task $\mathcal{T}_j \sim P(\mathcal{T})$, we can achieve good generalization performance (i.e., low test error) by using the adaptation procedure with the meta-learned parameters as $\boldsymbol{\phi}_j = \mathcal{A}lg(\boldsymbol{\theta}_{\mathrm{ML}}^*, \mathcal{D}_j^{\mathrm{tr}})$.

In the case of MAML [15], $\mathcal{A}lg(\boldsymbol{\theta}, \mathcal{D})$ corresponds to one or multiple steps of gradient descent initialized at $\boldsymbol{\theta}$. For example, if one step of gradient descent is used, we have:

$$\boldsymbol{\phi}_i \equiv \mathcal{A}lg(\boldsymbol{\theta}, \mathcal{D}_i^{\mathrm{tr}}) = \boldsymbol{\theta} - \alpha \nabla_{\boldsymbol{\theta}} \mathcal{L}(\boldsymbol{\theta}, \mathcal{D}_i^{\mathrm{tr}}). \quad \text{(inner-level of MAML)} \tag{2}$$

Typically, $\alpha$ is a scalar hyperparameter, but can also be a learned vector [34]. Hence, for MAML, the meta-learned parameter ($\boldsymbol{\theta}_{\mathrm{ML}}^*$) has a learned inductive bias that is particularly well-suited for fine-tuning on tasks from $P(\mathcal{T})$ using $K$ samples. To solve the outer-level problem with gradient-based methods, we require a way to differentiate through $\mathcal{A}lg$. In the case of MAML, this corresponds to backpropagating through the dynamics of gradient descent.

## 2.2 Proximal Regularization in the Inner Level

To have sufficient learning in the inner level while also avoiding over-fitting, $\mathcal{A}lg$ needs to incorporate some form of regularization. Since MAML uses a small number of gradient steps, this corresponds to early stopping and can be interpreted as a form of regularization and Bayesian prior [20]. In cases like ill-conditioned optimization landscapes and medium-shot learning, we may want to take many gradient steps, which poses two challenges for MAML. First, we need to store and differentiate through the long optimization path of $\mathcal{A}lg$, which imposes a considerable computation and memory burden. Second, the dependence of the model-parameters $\{\boldsymbol{\phi}_i\}$ on the meta-parameters ($\boldsymbol{\theta}$) shrinks and vanishes as the number of gradient steps in $\mathcal{A}lg$ grows, making meta-learning difficult. To overcome these limitations, we consider a more explicitly regularized algorithm:

$$\mathcal{A}lg^{\star}(\boldsymbol{\theta}, \mathcal{D}_i^{\mathrm{tr}}) = \operatorname*{argmin}_{\boldsymbol{\phi}' \in \Phi} \mathcal{L}(\boldsymbol{\phi}', \mathcal{D}_i^{\mathrm{tr}}) + \frac{\lambda}{2} \|\boldsymbol{\phi}' - \boldsymbol{\theta}\|^2. \tag{3}$$

The proximal regularization term in Eq. 3 encourages $\boldsymbol{\phi}_i$ to remain close to $\boldsymbol{\theta}$, thereby retaining a strong dependence throughout. The regularization strength ($\lambda$) plays a role similar to the learning rate ($\alpha$) in MAML, controlling the strength of the prior ($\boldsymbol{\theta}$) relative to the data ($\mathcal{D}_{\mathcal{T}}^{\mathrm{tr}}$). Like $\alpha$, the regularization strength $\lambda$ may also be learned. Furthermore, both $\alpha$ and $\lambda$ can be scalars, vectors, or full matrices. For simplicity, we treat $\lambda$ as a scalar hyperparameter. In Eq. 3, we use $\star$ to denote that the optimization problem is solved exactly. In practice, we use iterative algorithms (denoted by $\mathcal{A}lg$) for finite iterations, which return approximate minimizers. We explicitly consider the discrepancy between approximate and exact solutions in our analysis.

## 2.3 The Bi-Level Optimization Problem

For notation convenience, we will sometimes express the dependence on task $\mathcal{T}_i$ using a subscript instead of arguments, e.g. we write:

$$\mathcal{L}_i(\boldsymbol{\phi}) := \mathcal{L}(\boldsymbol{\phi}, \mathcal{D}_i^{\mathrm{test}}), \quad \hat{\mathcal{L}}_i(\boldsymbol{\phi}) := \mathcal{L}(\boldsymbol{\phi}, \mathcal{D}_i^{\mathrm{tr}}), \quad \mathcal{A}lg_i(\boldsymbol{\theta}) := \mathcal{A}lg(\boldsymbol{\theta}, \mathcal{D}_i^{\mathrm{tr}}).$$

With this notation, the bi-level meta-learning problem can be written more generally as:

$$\begin{aligned} \boldsymbol{\theta}_{\mathrm{ML}}^* &:= \operatorname*{argmin}_{\boldsymbol{\theta} \in \Theta} F(\boldsymbol{\theta}), \text{ where } F(\boldsymbol{\theta}) = \frac{1}{M} \sum_{i=1}^{M} \mathcal{L}_i\big(\mathcal{A}lg_i^{\star}(\boldsymbol{\theta})\big), \text{ and} \\ \mathcal{A}lg_i^{\star}(\boldsymbol{\theta}) &:= \operatorname*{argmin}_{\boldsymbol{\phi}' \in \Phi} G_i(\boldsymbol{\phi}', \boldsymbol{\theta}), \text{ where } G_i(\boldsymbol{\phi}', \boldsymbol{\theta}) = \hat{\mathcal{L}}_i(\boldsymbol{\phi}') + \frac{\lambda}{2} \|\boldsymbol{\phi}' - \boldsymbol{\theta}\|^2. \end{aligned} \tag{4}$$

## 2.4 Total and Partial Derivatives

We use $\boldsymbol{d}$ to denote the total derivative and $\nabla$ to denote partial derivative. For nested function of the form $\mathcal{L}_i(\boldsymbol{\phi}_i)$ where $\boldsymbol{\phi}_i = \mathcal{A}lg_i(\boldsymbol{\theta})$, we have from chain rule

$$\boldsymbol{d}_{\boldsymbol{\theta}} \mathcal{L}_i(\mathcal{A}lg_i(\boldsymbol{\theta})) = \frac{d\mathcal{A}lg_i(\boldsymbol{\theta})}{d\boldsymbol{\theta}} \nabla_{\boldsymbol{\phi}} \mathcal{L}_i(\boldsymbol{\phi}) \mid_{\boldsymbol{\phi} = \mathcal{A}lg_i(\boldsymbol{\theta})} = \frac{d\mathcal{A}lg_i(\boldsymbol{\theta})}{d\boldsymbol{\theta}} \nabla_{\boldsymbol{\phi}} \mathcal{L}_i(\mathcal{A}lg_i(\boldsymbol{\theta}))$$

Note the important distinction between $\boldsymbol{d}_{\boldsymbol{\theta}} \mathcal{L}_i(\mathcal{A}lg_i(\boldsymbol{\theta}))$ and $\nabla_{\boldsymbol{\phi}} \mathcal{L}_i(\mathcal{A}lg_i(\boldsymbol{\theta}))$. The former passes derivatives through $\mathcal{A}lg_i(\boldsymbol{\theta})$ while the latter does not. $\nabla_{\boldsymbol{\phi}} \mathcal{L}_i(\mathcal{A}lg_i(\boldsymbol{\theta}))$ is simply the gradient function, i.e. $\nabla_{\boldsymbol{\phi}} \mathcal{L}_i(\boldsymbol{\phi})$, evaluated at $\boldsymbol{\phi} = \mathcal{A}lg_i(\boldsymbol{\theta})$. Also note that $\boldsymbol{d}_{\boldsymbol{\theta}} \mathcal{L}_i(\mathcal{A}lg_i(\boldsymbol{\theta}))$ and $\nabla_{\boldsymbol{\phi}} \mathcal{L}_i(\mathcal{A}lg_i(\boldsymbol{\theta}))$ are $d$–dimensional vectors, while $\frac{d\mathcal{A}lg_i(\boldsymbol{\theta})}{d\boldsymbol{\theta}}$ is a $(d \times d)$–size Jacobian matrix. Throughout this text, we will also use $\boldsymbol{d}_{\boldsymbol{\theta}}$ and $\frac{d}{d\boldsymbol{\theta}}$ interchangeably.

## 3 The Implicit MAML Algorithm

Our aim is to solve the bi-level meta-learning problem in Eq. 4 using an iterative gradient based algorithm of the form $\boldsymbol{\theta} \leftarrow \boldsymbol{\theta} - \eta \, \boldsymbol{d}_{\boldsymbol{\theta}} F(\boldsymbol{\theta})$. Although we derive our method based on standard gradient descent for simplicity, any other optimization method, such as quasi-Newton or Newton methods, Adam [28], or gradient descent with momentum can also be used without modification. The gradient descent update be expanded using the chain rule as

$$\boldsymbol{\theta} \leftarrow \boldsymbol{\theta} - \eta \, \frac{1}{M} \sum_{i=1}^{M} \frac{d\mathcal{A}lg_i^{\star}(\boldsymbol{\theta})}{d\boldsymbol{\theta}} \, \nabla_{\boldsymbol{\phi}} \mathcal{L}_i(\mathcal{A}lg_i^{\star}(\boldsymbol{\theta})). \tag{5}$$

Here, $\nabla_{\boldsymbol{\phi}} \mathcal{L}_i(\mathcal{A}lg_i^{\star}(\boldsymbol{\theta}))$ is simply $\nabla_{\boldsymbol{\phi}} \mathcal{L}_i(\boldsymbol{\phi}) \mid_{\boldsymbol{\phi} = \mathcal{A}lg_i^{\star}(\boldsymbol{\theta})}$ which can be easily obtained in practice via automatic differentiation. For this update rule, we must compute $\frac{d\mathcal{A}lg_i^{\star}(\boldsymbol{\theta})}{d\boldsymbol{\theta}}$, where $\mathcal{A}lg_i^{\star}$ is implicitly defined as an optimization problem (Eq. 4), which presents the primary challenge. We now present an efficient algorithm (in compute and memory) to compute the meta-gradient..

### 3.1 Meta-Gradient Computation

If $\mathcal{A}lg_i^{\star}(\boldsymbol{\theta})$ is implemented as an iterative algorithm, such as gradient descent, then one way to compute $\frac{d\mathcal{A}lg_i^{\star}(\boldsymbol{\theta})}{d\boldsymbol{\theta}}$ is to propagate derivatives through the iterative process, either in forward mode or reverse mode. However, this has the drawback of depending explicitly on the path of the optimization, which has to be fully stored in memory, quickly becoming intractable when the number of gradient steps needed is large. Furthermore, for second order optimization methods, such as Newton's method, third derivatives are needed which are difficult to obtain. Furthermore, this approach becomes impossible when non-differentiable operations, such as line-searches, are used. However, by recognizing that $\mathcal{A}lg_i^{\star}$ is implicitly defined as the solution to an optimization problem, we may employ a different strategy that does not need to consider the path of the optimization but only the final result. This is derived in the following Lemma.

**Lemma 1.** *(Implicit Jacobian) Consider $\mathcal{A}lg_i^{\star}(\boldsymbol{\theta})$ as defined in Eq. 4 for task $\mathcal{T}_i$. Let $\boldsymbol{\phi}_i = \mathcal{A}lg_i^{\star}(\boldsymbol{\theta})$ be the result of $\mathcal{A}lg_i^{\star}(\boldsymbol{\theta})$. If $\left( \boldsymbol{I} + \frac{1}{\lambda} \nabla_{\boldsymbol{\phi}}^2 \hat{\mathcal{L}}_i(\boldsymbol{\phi}_i) \right)$ is invertible, then the derivative Jacobian is*

$$\frac{d\mathcal{A}lg_i^{\star}(\boldsymbol{\theta})}{d\boldsymbol{\theta}} = \left( \boldsymbol{I} + \frac{1}{\lambda} \, \nabla_{\boldsymbol{\phi}}^2 \hat{\mathcal{L}}_i(\boldsymbol{\phi}_i) \right)^{-1}. \tag{6}$$

Note that the derivative (Jacobian) depends only on the final result of the algorithm, and not the path taken by the algorithm. Thus, in principle any approach of algorithm can be used to compute $\mathcal{A}lg_i^{\star}(\boldsymbol{\theta})$, thereby decoupling meta-gradient computation from choice of inner level optimizer.

**Practical Algorithm:** While Lemma 1 provides an idealized way to compute the $\mathcal{A}lg_i^{\star}$ Jacobians and thus by extension the meta-gradient, it may be difficult to directly use it in practice. Two issues are particularly relevant. First, the meta-gradients require computation of $\mathcal{A}lg_i^{\star}(\boldsymbol{\theta})$, which is the exact solution to the inner optimization problem. In practice, we may be able to obtain only approximate solutions. Second, explicitly forming and inverting the matrix in Eq. 6 for computing

---

**Algorithm 1** Implicit Model-Agnostic Meta-Learning (iMAML)

---

1: **Require:** Distribution over tasks $P(\mathcal{T})$, outer step size $\eta$, regularization strength $\lambda$,
2: **while** not converged **do**
3:   Sample mini-batch of tasks $\{\mathcal{T}_i\}_{i=1}^{B} \sim P(\mathcal{T})$
4:   **for** Each task $\mathcal{T}_i$ **do**
5:     Compute task meta-gradient $\boldsymbol{g}_i = \text{Implicit-Meta-Gradient}(\mathcal{T}_i, \boldsymbol{\theta}, \lambda)$
6:   **end for**
7:   Average above gradients to get $\hat{\nabla}F(\boldsymbol{\theta}) = (1/B)\sum_{i=1}^{B} \boldsymbol{g}_i$
8:   Update meta-parameters with gradient descent: $\boldsymbol{\theta} \leftarrow \boldsymbol{\theta} - \eta\hat{\nabla}F(\boldsymbol{\theta})$   // *(or Adam)*
9: **end while**

---

---

**Algorithm 2** Implicit Meta-Gradient Computation

---

1: **Input:** Task $\mathcal{T}_i$, meta-parameters $\boldsymbol{\theta}$, regularization strength $\lambda$
2: **Hyperparameters:** Optimization accuracy thresholds $\delta$ and $\delta'$
3: Obtain task parameters $\boldsymbol{\phi}_i$ using iterative optimization solver such that: $\|\boldsymbol{\phi}_i - \mathcal{A}lg_i^\star(\boldsymbol{\theta})\| \leq \delta$
4: Compute partial outer-level gradient $\boldsymbol{v}_i = \nabla_\phi \mathcal{L}_\mathcal{T}(\boldsymbol{\phi}_i)$
5: Use an iterative solver (e.g. CG) along with reverse mode differentiation (to compute Hessian vector products) to compute $\boldsymbol{g}_i$ such that: $\|\boldsymbol{g}_i - \left(\boldsymbol{I} + \frac{1}{\lambda}\nabla^2\hat{\mathcal{L}}_i(\boldsymbol{\phi}_i)\right)^{-1}\boldsymbol{v}_i\| \leq \delta'$
6: **Return:** $\boldsymbol{g}_i$

---

the Jacobian may be intractable for large deep neural networks. To address these difficulties, we consider approximations to the idealized approach that enable a practical algorithm.

First, we consider an approximate solution to the inner optimization problem, that can be obtained with iterative optimization algorithms like gradient descent.

**Definition 1.** *($\delta$–approx. algorithm) Let $\mathcal{A}lg_i(\boldsymbol{\theta})$ be a $\delta$–accurate approximation of $\mathcal{A}lg_i^\star(\boldsymbol{\theta})$, i.e.*

$$\|\mathcal{A}lg_i(\boldsymbol{\theta}) - \mathcal{A}lg_i^\star(\boldsymbol{\theta})\| \leq \delta$$

Second, we will perform a partial or approximate matrix inversion given by:

**Definition 2.** *($\delta'$–approximate Jacobian-vector product) Let $\boldsymbol{g}_i$ be a vector such that*

$$\|\boldsymbol{g}_i - \left(\boldsymbol{I} + \frac{1}{\lambda}\nabla_\phi^2\hat{\mathcal{L}}_i(\boldsymbol{\phi}_i)\right)^{-1}\nabla_\phi\mathcal{L}_i(\boldsymbol{\phi}_i)\| \leq \delta'$$

*where $\boldsymbol{\phi}_i = \mathcal{A}lg_i(\boldsymbol{\theta})$ and $\mathcal{A}lg_i$ is based on definition 1.*

Note that $\boldsymbol{g}_i$ in definition 2 is an approximation of the meta-gradient for task $\mathcal{T}_i$. Observe that $\boldsymbol{g}_i$ can be obtained as an approximate solution to the optimization problem:

$$\min_{\boldsymbol{w}} \quad \frac{1}{2}\boldsymbol{w}^\top\left(\boldsymbol{I} + \frac{1}{\lambda}\nabla_\phi^2\hat{\mathcal{L}}_i(\boldsymbol{\phi}_i)\right)\boldsymbol{w} - \boldsymbol{w}^\top\nabla_\phi\mathcal{L}_i(\boldsymbol{\phi}_i) \tag{7}$$

The conjugate gradient (CG) algorithm is particularly well suited for this problem due to its excellent iteration complexity and requirement of only Hessian-vector products of the form $\nabla^2\hat{\mathcal{L}}_i(\boldsymbol{\phi}_i)\boldsymbol{v}$. Such hessian-vector products can be obtained cheaply without explicitly forming or storing the Hessian matrix (as we discuss in Appendix C). This CG based inversion has been successfully deployed in Hessian-free or Newton-CG methods for deep learning [36, 44] and trust region methods in reinforcement learning [52, 47]. Algorithm 1 presents the full practical algorithm. Note that these approximations to develop a practical algorithm introduce errors in the meta-gradient computation. We analyze the impact of these errors in Section 3.2 and show that they are controllable. See Appendix A for how iMAML generalizes prior gradient optimization based meta-learning algorithms.

## 3.2 Theory

In Section 3.1, we outlined a practical algorithm that makes approximations to the idealized update rule of Eq. 5. Here, we attempt to analyze the impact of these approximations, and also understand the computation and memory requirements of iMAML. We find that iMAML can match the

Table 1: Compute and memory for computing the meta-gradient when using a $\delta$–accurate $\mathcal{A}lg_i$, and the corresponding approximation error. Our compute time is measured in terms of the number of $\nabla\hat{\mathcal{L}}_i$ computations. All results are in $\tilde{O}(\cdot)$ notation, which hide additional log factors; the error bound hides additional problem dependent Lipshitz and smoothness parameters (see the respective Theorem statements). $\kappa \geq 1$ is the condition number for inner objective $G_i$ (see Equation 4), and $D$ is the diameter of the search space. The notions of error are subtly different: we assume all methods solve the inner optimization to error level of $\delta$ (as per definition 1). For our algorithm, the error refers to the $\ell_2$ error in the computation of $d_{\boldsymbol{\theta}}\mathcal{L}_i(\mathcal{A}lg_i^\star(\boldsymbol{\theta}))$. For the other algorithms, the error refers to the $\ell_2$ error in the computation of $d_{\boldsymbol{\theta}}\mathcal{L}_i(\mathcal{A}lg_i(\boldsymbol{\theta}))$. We use Prop 3.1 of Shaban et al. [53] to provide the guarantee we use. See Appendix D for additional discussion.

| Algorithm | Compute | Memory | Error |
|---|---|---|---|
| MAML (GD + full back-prop) | $\kappa \log\left(\frac{D}{\delta}\right)$ | $\text{Mem}(\nabla\hat{\mathcal{L}}_i) \cdot \kappa \, \log\left(\frac{D}{\delta}\right)$ | 0 |
| MAML (Nesterov's AGD + full back-prop) | $\sqrt{\kappa} \log\left(\frac{D}{\delta}\right)$ | $\text{Mem}(\nabla\hat{\mathcal{L}}_i) \cdot \sqrt{\kappa} \, \log\left(\frac{D}{\delta}\right)$ | 0 |
| Truncated back-prop [53] (GD) | $\kappa \log\left(\frac{D}{\delta}\right)$ | $\text{Mem}(\nabla\hat{\mathcal{L}}_i) \cdot \kappa \, \log\left(\frac{1}{\epsilon}\right)$ | $\epsilon$ |
| Implicit MAML (this work) | $\sqrt{\kappa} \log\left(\frac{D}{\delta}\right)$ | $\text{Mem}(\nabla\hat{\mathcal{L}}_i)$ | $\delta$ |

minimax computational complexity of backpropagating through the path of the inner optimizer, but is substantially better in terms of memory usage. This work to our knowledge also provides the first non-asymptotic result that analyzes approximation error due to implicit gradients. Theorem 1 provides the computational and memory complexity for obtaining an $\epsilon$–approximate meta-gradient. We assume $\mathcal{L}_i$ is smooth but do not require it to be convex. We assume that $G_i$ in Eq. 4 is strongly convex, which can be made possible by appropriate choice of $\lambda$. The key to our analysis is a second order Lipshitz assumption, i.e. $\hat{\mathcal{L}}_i(\cdot)$ is $\rho$-Lipshitz Hessian. This assumption and setting has received considerable attention in recent optimization and deep learning literature [26, 42].

Table 1 summarizes our complexity results and compares with MAML and truncated backpropagation [53] through the path of the inner optimizer. We use $\kappa$ to denote the condition number of the inner problem induced by $G_i$ (see Equation 4), which can be viewed as a measure of hardness of the inner optimization problem. $\text{Mem}(\nabla\hat{\mathcal{L}}_i)$ is the memory taken to compute a single derivative $\nabla\hat{\mathcal{L}}_i$. Under the assumption that Hessian vector products are computed with the reverse mode of autodifferentiation, we will have that both: the compute time and memory used for computing a Hessian vector product are with a (universal) constant factor of the compute time and memory used for computing $\nabla\hat{\mathcal{L}}_i$ itself (see Appendix C). This allows us to measure the compute time in terms of the number of $\nabla\hat{\mathcal{L}}_i$ computations. We refer readers to Appendix D for additional discussion about the algorithms and their trade-offs.

Our main theorem is as follows:

**Theorem 1.** *(Informal Statement; Approximation error in Algorithm 2) Suppose that: $\mathcal{L}_i(\cdot)$ is $B$ Lipshitz and $L$ smooth function; that $G_i(\cdot, \boldsymbol{\theta})$ (in Eq. 4) is a $\mu$-strongly convex function with condition number $\kappa$; that $D$ is the diameter of search space for $\phi$ in the inner optimization problem (i.e. $\|\mathcal{A}lg_i^\star(\boldsymbol{\theta})\| \leq D$); and $\hat{\mathcal{L}}_i(\cdot)$ is $\rho$-Lipshitz Hessian.*

*Let $\boldsymbol{g}_i$ be the task meta-gradient returned by Algorithm 2. For any task $i$ and desired accuracy level $\epsilon$, Algorithm 2 computes an approximate task-specific meta-gradient with the following guarantee:*

$$\|\boldsymbol{g}_i - d_{\boldsymbol{\theta}}\mathcal{L}_i(\mathcal{A}lg_i^\star(\boldsymbol{\theta}))\| \leq \epsilon \,.$$

*Furthermore, under the assumption that the Hessian vector products are computed by the reverse mode of autodifferentiation (Assumption 1), Algorithm 2 can be implemented using at most $\tilde{O}\left(\sqrt{\kappa} \log\left(\frac{poly(\kappa, D, B, L, \rho, \mu, \lambda)}{\epsilon}\right)\right)$ gradient computations of $\hat{\mathcal{L}}_i(\cdot)$ and $2 \cdot \text{Mem}(\nabla\hat{\mathcal{L}}_i)$ memory.*

The formal statement of the theorem and the proof are provided the appendix. Importantly, the algorithm's memory requirement is equivalent to the memory needed for Hessian-vector products which is a small constant factor over the memory required for gradient computations, assuming the reverse mode of auto-differentiation is used. Finally, based on the above, we also present corollary 1 in the appendix which shows that iMAML efficiently finds a stationary point of $F(\cdot)$, due to iMAML having controllable exact-solve error.

## 4  Experimental Results and Discussion

In our experimental evaluation, we aim to answer the following questions empirically: (1) Does the iMAML algorithm asymptotically compute the exact meta-gradient? (2) With finite iterations, does iMAML approximate the meta-gradient more accurately compared to MAML? (3) How does the computation and memory requirements of iMAML compare with MAML? (4) Does iMAML lead to better results in realistic meta-learning problems? We have answered (1) - (3) through our theoretical analysis, and now attempt to validate it through numerical simulations. For (1) and (2), we will use a simple synthetic example for which we can compute the exact meta-gradient and compare against it (exact-solve error, see definition 3). For (3) and (4), we will use the common few-shot image recognition domains of Omniglot and Mini-ImageNet.

To study the question of meta-gradient accuracy, Figure 2 considers a synthetic regression example, where the predictions are linear in parameters. This provides an analytical expression for $\mathcal{A}lg_i^\star$ allowing us to compute the true meta-gradient. We fix gradient descent (GD) to be the inner optimizer for both MAML and iMAML. The problem is constructed so that the condition number $(\kappa)$ is large, thereby necessitating many GD steps. We find that both iMAML and MAML asymptotically match the exact meta-gradient, but iMAML computes a better approximation in finite iterations. We observe that with 2 CG iterations, iMAML incurs a small terminal error. This is consistent with our theoretical analysis. In Algorithm 2, $\delta$ is dominated by $\delta'$ when only a small number of CG steps are used. However, the terminal error vanishes with just 5 CG steps. The computational cost of 1 CG step is comparable to 1 inner GD step with the MAML algorithm, since both require 1 hessian-vector product (see section C for discussion). Thus, the computational cost as well as memory of iMAML with 100 inner GD steps is significantly smaller than MAML with 100 GD steps.

To study (3), we turn to the Omniglot dataset [30] which is a popular few-shot image recognition domain. Figure 2 presents compute and memory trade-off for MAML and iMAML (on 20-way, 5-shot Omniglot). Memory for iMAML is based on Hessian-vector products and is independent of the number of GD steps in the inner loop. The memory use is also independent of the number of CG iterations, since the intermediate computations need not be stored in memory. On the other hand, memory for MAML grows linearly in grad steps, reaching the capacity of a 12 GB GPU in approximately 16 steps. First-order MAML (FOMAML) does not back-propagate through the optimization process, and thus the computational cost is only that of performing gradient descent, which is needed for all the algorithms. The computational cost for iMAML is also similar to FOMAML along with a constant overhead for CG that depends on the number of CG steps. Note however, that FOMAML does not compute an accurate meta-gradient, since it ignores the Jacobian. Compared to FOMAML, the compute cost of MAML grows at a faster rate. FOMAML requires only gradient computations, while backpropagating through GD (as done in MAML) requires a Hessian-vector products at each iteration, which are more expensive.

Finally, we study empirical performance of iMAML on the Omniglot and Mini-ImageNet domains. Following the few-shot learning protocol in prior work [57], we run the iMAML algorithm on the

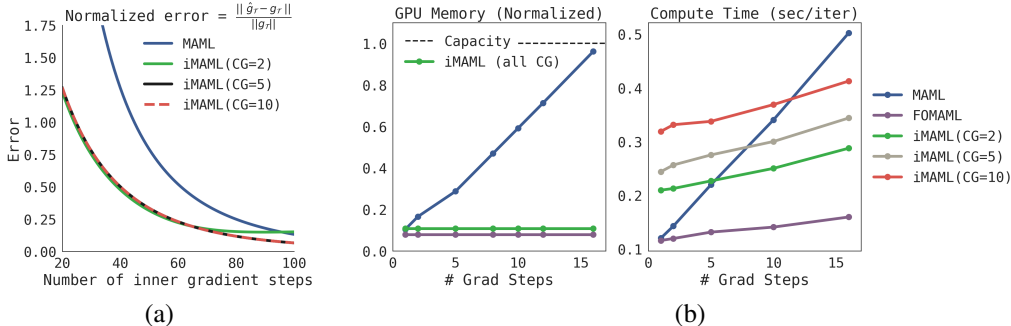

Figure 2: Accuracy, Computation, and Memory tradeoffs of iMAML, MAML, and FOMAML. (a) Meta-gradient accuracy level in synthetic example. Computed gradients are compared against the exact meta-gradient per Def 3. (b) Computation and memory trade-offs with 4 layer CNN on 20-way-5-shot Omniglot task. We implemented iMAML in PyTorch, and for an apples-to-apples comparison, we use a PyTorch implementation of MAML from: https://github.com/dragen1860/MAML-Pytorch

Table 2: Omniglot results. MAML results are taken from the original work of Finn et al. [15], and first-order MAML and Reptile results are from Nichol et al. [43]. iMAML with gradient descent (GD) uses 16 and 25 steps for 5-way and 20-way tasks respectively. iMAML with Hessian-free uses 5 CG steps to compute the search direction and performs line-search to pick step size. Both versions of iMAML use $\lambda = 2.0$ for regularization, and 5 CG steps to compute the task meta-gradient.

| Algorithm | 5-way 1-shot | 5-way 5-shot | 20-way 1-shot | 20-way 5-shot |
|---|---|---|---|---|
| MAML [15] | $98.7 \pm 0.4\%$ | $\mathbf{99.9 \pm 0.1\%}$ | $95.8 \pm 0.3\%$ | $98.9 \pm 0.2\%$ |
| first-order MAML [15] | $98.3 \pm 0.5\%$ | $99.2 \pm 0.2\%$ | $89.4 \pm 0.5\%$ | $97.9 \pm 0.1\%$ |
| Reptile [43] | $97.68 \pm 0.04\%$ | $99.48 \pm 0.06\%$ | $89.43 \pm 0.14\%$ | $97.12 \pm 0.32\%$ |
| iMAML, GD (ours) | $99.16 \pm 0.35\%$ | $99.67 \pm 0.12\%$ | $94.46 \pm 0.42\%$ | $98.69 \pm 0.1\%$ |
| iMAML, Hessian-Free (ours) | $\mathbf{99.50 \pm 0.26\%}$ | $99.74 \pm 0.11\%$ | $\mathbf{96.18 \pm 0.36\%}$ | $\mathbf{99.14 \pm 0.1\%}$ |

dataset for different numbers of class labels and shots (in the N-way, K-shot setting), and compare two variants of iMAML with published results of the most closely related algorithms: MAML, FOMAML, and Reptile. While these methods are not state-of-the-art on this benchmark, they provide an apples-to-apples comparison for studying the use of implicit gradients in optimization-based meta-learning. For a fair comparison, we use the identical convolutional architecture as these prior works. Note however that architecture tuning can lead to better results for all algorithms [27].

The first variant of iMAML we consider involves solving the inner level problem (the regularized objective function in Eq. 4) using gradient descent. The meta-gradient is computed using conjugate gradient, and the meta-parameters are updated using Adam. This presents the most straightforward comparison with MAML, which would follow a similar procedure, but backpropagate through the path of optimization as opposed to invoking implicit differentiation. The second variant of iMAML uses a second order method for the inner level problem. In particular, we consider the Hessian-free or Newton-CG [44, 36] method. This method makes a local quadratic approximation to the objective function (in our case, $G(\phi', \boldsymbol{\theta})$ and approximately computes the Newton search direction using CG. Since CG requires only Hessian-vector products, this way of approximating the Newton search direction is scalable to large deep neural networks. The step size can be computed using regularization, damping, trust-region, or linesearch. We use a linesearch on the training loss in our experiments to also illustrate how our method can handle non-differentiable inner optimization loops. We refer the readers to Nocedal & Wright [44] and Martens [36] for a more detailed exposition of this optimization algorithm. Similar approaches have also gained prominence in reinforcement learning [52, 47].

Tables 2 and 3 present the results on Omniglot and Mini-ImageNet, respectively. On the Omniglot domain, we find that the GD version of iMAML is competitive with the full MAML algorithm, and substatially better than its approximations (i.e., first-order MAML and Reptile), especially for the harder 20-way tasks. We also find that iMAML with Hessian-free optimization performs substantially better than the other methods, suggesting that powerful optimizers in the inner loop can offer benefits to meta-learning. In the Mini-ImageNet domain,

Table 3: Mini-ImageNet 5-way-1-shot accuracy

| Algorithm | 5-way 1-shot |
|---|---|
| MAML | $48.70 \pm 1.84\ \%$ |
| first-order MAML | $48.07 \pm 1.75\ \%$ |
| Reptile | $49.97 \pm 0.32\ \%$ |
| iMAML GD (ours) | $48.96 \pm 1.84\ \%$ |
| iMAML HF (ours) | $49.30 \pm 1.88\ \%$ |

we find that iMAML performs better than MAML and FOMAML. We used $\lambda = 0.5$ and 10 gradient steps in the inner loop. We did not perform an extensive hyperparameter sweep, and expect that the results can improve with better hyperparameters. 5 CG steps were used to compute the meta-gradient. The Hessian-free version also uses 5 CG steps for the search direction. Additional experimental details are Appendix F.

## 5 Related Work

Our work considers the general meta-learning problem [51, 55, 41], including few-shot learning [30, 57]. Meta-learning approaches can generally be categorized into metric-learning approaches that learn an embedding space where non-parametric nearest neighbors works well [29, 57, 54, 45, 3], black-box approaches that train a recurrent or recursive neural network to take datapoints as input

and produce weight updates [25, 5, 33, 48] or predictions for new inputs [50, 12, 58, 40, 38], and optimization-based approaches that use bi-level optimization to embed learning procedures, such as gradient descent, into the meta-optimization problem [15, 13, 8, 60, 34, 17, 59, 23]. Hybrid approaches have also been considered to combine the benefits of different approaches [49, 56]. We build upon optimization-based approaches, particularly the MAML algorithm [15], which meta-learns an initial set of parameters such that gradient-based fine-tuning leads to good generalization. Prior work has considered a number of inner loops, ranging from a very general setting where all parameters are adapted using gradient descent [15], to more structured and specialized settings, such as ridge regression [8], Bayesian linear regression [23], and simulated annealing [2]. The main difference between our work and these approaches is that we show how to analytically derive the gradient of the outer objective without differentiating through the inner learning procedure.

Mathematically, we view optimization-based meta-learning as a bi-level optimization problem. Such problems have been studied in the context of few-shot meta-learning (as discussed previously), gradient-based hyperparameter optimization [35, 46, 19, 11, 10], and a range of other settings [4, 31]. Some prior works have derived implicit gradients for related problems [46, 11, 4] while others propose innovations to aid back-propagation through the optimization path for specific algorithms [35, 19, 24], or approximations like truncation [53]. While the broad idea of implicit differentiation is well known, it has not been empirically demonstrated in the past for learning more than a few parameters (e.g., hyperparameters), or highly structured settings such as quadratic programs [4]. In contrast, our method meta-trains deep neural networks with thousands of parameters. Closest to our setting is the recent work of Lee et al. [32], which uses implicit differentiation for quadratic programs in a final SVM layer. In contrast, our formulation allows for adapting the full network for generic objectives (beyond hinge-loss), thereby allowing for wider applications.

We also note that prior works involving implicit differentiation make a strong assumption of an exact solution in the inner level, thereby providing only asymptotic guarantees. In contrast, we provide finite time guarantees which allows us to analyze the case where the inner level is solved approximately. In practice, the inner level is likely to be solved using iterative optimization algorithms like gradient descent, which only return approximate solutions with finite iterations. Thus, this paper places implicit gradient methods under a strong theoretical footing for practically use.

## 6   Conclusion

In this paper, we develop a method for optimization-based meta-learning that removes the need for differentiating through the inner optimization path, allowing us to decouple the outer meta-gradient computation from the choice of inner optimization algorithm. We showed how this gives us significant gains in compute and memory efficiency, and also conceptually allows us to use a variety of inner optimization methods. While we focused on developing the foundations and theoretical analysis of this method, we believe that this work opens up a number of interesting avenues for future study.

**Broader classes of inner loop procedures.** While we studied different gradient-based optimization methods in the inner loop, iMAML can in principle be used with a variety of inner loop algorithms, including dynamic programming methods such as $Q$-learning, two-player adversarial games such as GANs, energy-based models [39], and actor-critic RL methods, and higher-order model-based trajectory optimization methods. This significantly expands the kinds of problems that optimization-based meta-learning can be applied to.

**More flexible regularizers.** We explored one very simple regularization, $\ell_2$ regularization to the parameter initialization, which already increases the expressive power over the implicit regularization that MAML provides through truncated gradient descent. To further allow the model to flexibly regularize the inner optimization, a simple extension of iMAML is to learn a vector- or matrix-valued $\lambda$, which would enable the meta-learner model to co-adapt and co-regularize various parameters of the model. Regularizers that act on parameterized density functions would also enable meta-learning to be effective for few-shot density estimation.

## Acknowledgements

Aravind Rajeswaran thanks Emo Todorov for valuable discussions about implicit gradients and potential application domains; Aravind Rajeswaran also thanks Igor Mordatch and Rahul Kidambi for helpful discussions and feedback. Sham Kakade acknowledges funding from the Washington Research Foundation for innovation in Data-intensive Discovery; Sham Kakade also graciously acknowledges support from ONR award N00014-18-1-2247, NSF Award CCF-1703574, and NSF CCF 1740551 award.

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
