[Supplementary Material]

## A    Relationship between iMAML and Prior Algorithms

The presented iMAML algorithm has close connections, as well as notable differences, to a number of related algorithms like MAML [15], first-order MAML, and Reptile [43]. Conventionally, these algorithms do not consider any explicit regularization in the inner-level and instead rely on early stopping, through only a few gradient descent steps. In our problem setting described in Eq. 4, we consider an explicitly regularized inner-level problem (refer to discussion in Section 2.2). We describe the connections between the algorithms in this explicitly regularized setting below.

**MAML**. The MAML algorithm first invokes an iterative algorithm to solve the inner optimization problem (see definition 1). Subsequently, it backpropagates through the path of the optimization algorithm to update the meta-parameters as:

$$\boldsymbol{\theta}^{k+1} = \boldsymbol{\theta}^k - \eta \, \frac{1}{M} \sum_{i=1}^{M} \boldsymbol{d_\theta} \mathcal{L}_i(\mathcal{A}lg_i(\boldsymbol{\theta}^k)).$$

Since $\mathcal{A}lg_i(\boldsymbol{\theta})$ approximates $\mathcal{A}lg_i^\star(\boldsymbol{\theta})$, it can be viewed that both MAML and iMAML intend to perform the same idealized update in Eq. 5. However, they perform the meta-gradient computation very differently. MAML backpropagates through the path of an iterative algorithm, while iMAML computes the meta-gradient through the implicit Jacobian approach outlined in Section 3.1 (see Figure 1 for a visual depiction). As a result, iMAML can be vastly more efficient in memory while having lesser or comparable computational requirements. It also allows for higher order optimization methods and non-differentiable components.

**First-order MAML** ignores the effect of meta-parameters $\boldsymbol{\theta}$ on task parameters $\{\boldsymbol{\phi}_i\}$ in the meta-gradient computation and updates the meta-parameters as:

$$\boldsymbol{\theta}^{k+1} = \boldsymbol{\theta}^k - \eta \, \frac{1}{M} \sum_{i=1}^{M} \nabla_{\boldsymbol{\phi}} \mathcal{L}_i(\boldsymbol{\phi}_i) \,|_{\boldsymbol{\phi}_i = \mathcal{A}lg_i(\boldsymbol{\theta}^k)}$$

Note that iMAML strictly generalizes this, since first-order MAML is simply iMAML when the conjugate gradient procedure is not invoked (or corresponds to 0 steps of CG). Thus, iMAML allows for an easy way to interpolate from first-order MAML to the full MAML algorithm.

**Reptile** [43], similar to first-order MAML, ignores the dependence of task-parameters on meta-parameters. However, instead of following the gradients at $\boldsymbol{\phi}_i = \mathcal{A}lg_i(\boldsymbol{\theta}^k)$, Reptile uses the task-parameters as targets and slowly moves meta-parameters towards them:

$$\boldsymbol{\theta}^{k+1} = \boldsymbol{\theta}^k - \eta \, \frac{1}{M} \sum_{i=1}^{M} (\boldsymbol{\theta}^k - \boldsymbol{\phi}_i).$$

From the proximal point equation in the proof of Lemma 1, we have $\boldsymbol{\phi}_i = \boldsymbol{\theta}^k - \frac{1}{\lambda} \nabla_{\boldsymbol{\phi}} \mathcal{L}_i(\boldsymbol{\phi}_i)$, using which we see that the Reptile equation becomes: $\boldsymbol{\theta}^{k+1} = \boldsymbol{\theta}^k - \frac{\eta}{\lambda M} \sum_{i=1}^{M} \nabla_{\boldsymbol{\phi}} \mathcal{L}_i(\boldsymbol{\phi}_i)$. Thus, Reptile and first-order MAML are identical in our problem formulation up to the choice of learning rate. Making the regularization explicit allows us to illustrate this equivalence.

## B    Optimization Preliminaries

Let $f : \mathbb{R}^d \to \mathbb{R}$. A function $f$ is $B$ Lipschitz (or $B$-bounded gradient norm) if for all $x \in \mathbb{R}^d$

$$||\nabla f(x)|| \le B \,.$$

Similarly, we say that a matrix valued function $M : \mathbb{R}^d \times \mathbb{R}^{d'} \to \mathbb{R}$ is $\rho$-Lipschitz if

$$||M(x) - M(x')|| \le \rho ||x - x'|| \,,$$

where $|| \cdot ||$ denotes the spectral norm.

We say that $f$ is $L$-smooth if for all $x, x' \in \mathbb{R}^d$

$$||\nabla f(x) - \nabla f(x')|| \le L ||x - x'||$$

and that $f$ is $\mu$-strongly convex if $f$ is convex and if for all $x, x' \in \mathbb{R}^d$,

$$\|\nabla f(x) - \nabla f(x')\| \geq \mu \|x - x'\|.$$

We will make use of the following black-box complexity of first-order gradient methods for minimizing strongly convex and smooth functions.

**Lemma 2.** *($\delta$-approximate solver; see [9]) Suppose $f$ is a function that is $L$-smooth and $\mu$ strongly convex. Define $\kappa := L/\mu$, and let $x^\star = \operatorname{argmin} f(x)$. Nesterov's accelerated gradient descent can be used to find a point $x$ such that:*

$$\|x - x^\star\| \leq \delta$$

*using a number of gradient computations of $f$ that is bounded as follows:*

$$\text{\# gradient computations of } f(\cdot) \leq 2\sqrt{\kappa} \, \log\left(2\kappa \frac{\|x^\star\|}{\delta}\right).$$

## C   Review: Time and Space Complexity of Hessian-Vector Products

We briefly discuss the time and space complexity of Hessian-vector product computation using the reverse mode of automatic differentiation. The reverse mode of automatic differentiation [6, 22] is the widely used method for automatic differentiation in modern software packages like TensorFlow and PyTorch [7]. Recall that for a differentiable function $f(x)$, the reverse mode of automatic differentiation computes $\nabla f(x)$ in time that is no more than a factor of 5 of the time it takes to compute $f(x)$ itself (see [22] for review). As our algorithm makes use of Hessian vector products, we will make use of the following assumption as to how Hessian vector products will be computed when executing Algorithm 2.

**Assumption 1.** *(Complexity of Hessian-vector product) We assume that the time to compute the Hessian-vector product $\nabla_\phi^2 \hat{\mathcal{L}}_i(\phi) v$ is no more than a (universal) constant over the time used to compute $\nabla \hat{\mathcal{L}}_i(\phi)$ (typically, this constant is 5). Furthermore, we assume that the memory used to compute the Hessian-vector product $\nabla_\phi^2 \hat{\mathcal{L}}_i(\phi) v$ is no more than twice the memory used when computing $\nabla \hat{\mathcal{L}}_i(\phi)$. This assumption is valid if the reverse mode of automatic differentiation is used to compute Hessian vector products (see [21]).*

A few remarks about this assumption are in order. With regards to computation, first observe that the gradient of the scalar function $\nabla_\phi \hat{\mathcal{L}}_i(\phi)^\top v$ is the desired Hessian vector product $\nabla_\phi^2 \hat{\mathcal{L}}_i(\phi) v$. Thus computing the Hessian vector product using the reverse mode is within a constant factor of computing the function itself, which is simply the cost of computing $\nabla \hat{\mathcal{L}}_i(\phi)^\top v$. The issue of memory is more subtle (see [21]), which we now discuss. The memory used to compute the gradient of a scalar cost function $f(x)$ using the reverse mode of auto-differentiation is proportional to the size of the computation graph; precisely, the memory required to compute the gradient is equal to the total space required to store all the intermediate variables used when computing $f(x)$. In practice, this is often much larger than the memory required to compute $f(x)$ itself, due to that all intermediate variables need not be simultaneously stored in memory when computing $f(x)$. However, for the special case of computing the gradient of the function $f(\phi) = \nabla_\phi \hat{\mathcal{L}}_i(\phi)^\top v$, the factor of 2 in the memory bound is a consequence of the following reason: first, using the reverse mode to compute $f(\phi)$ means we already have stored the computation graph of $\hat{\mathcal{L}}_i(\phi)$ itself. Furthermore, the size of the computation graph for computing $f(\phi) = \nabla_\phi \hat{\mathcal{L}}_i(\phi)^\top v$ is essentially the same size as the computation graph of $\hat{\mathcal{L}}_i(\phi)$. This leads to the factor of 2 memory bound; see Griewank [21] for further discussion.

## D   Additional Discussion About Compute and Memory Complexity

Our main complexity results are summarized in Table 1. For these results, we consider two notions of error that are subtly different, which we explicitly define below. Let $g_i$ be the computed meta-gradient for task $\mathcal{T}_i$. Then, the errors we consider are:

**Definition 3.** *Exact-solve error (our notion of error): Our goal is to accurately compute the gradient of $F(\theta)$ as defined in Equation 4, where $\mathcal{A}lg_i^\star(\theta)$ is an exact algorithm. Specifically, we seek to compute a $\boldsymbol{g}_i$ such that:*

$$\|\boldsymbol{g}_i - \boldsymbol{d_\theta}\mathcal{L}_i(\mathcal{A}lg_i^\star(\boldsymbol{\theta}))\| \le \epsilon$$

*where $\epsilon$ is the error in the gradient computation.*

**Definition 4.** *Approx-solve error: Here we suppose that $\mathcal{A}lg_i$ computes a $\delta$–accurate solution to the inner optimization problem over $G_i$ in Eq. 4, i.e. that $\mathcal{A}lg_i$ satisfies $\|\mathcal{A}lg_i(\boldsymbol{\theta}) - \mathcal{A}lg_i^\star(\boldsymbol{\theta})\| \le \delta$, as per definition 1. Then the objective is to compute a $\boldsymbol{g}$ such that:*

$$\|\boldsymbol{g} - \boldsymbol{d_\theta}\mathcal{L}_i(\mathcal{A}lg_i(\boldsymbol{\theta}))\| \le \epsilon$$

*where $\epsilon$ is the error in the gradient computation of $\boldsymbol{d_\theta}\mathcal{L}_i(\mathcal{A}lg_i(\boldsymbol{\theta}))$. Subtly, note that the gradient is with respect to the $\delta$-approximate algorithm, as opposed to using $\mathcal{A}lg_i^\star$.*

For the complexity results, we assume that MAML invokes $\mathcal{A}lg_i$ to get a $\delta$-approximate solution for inner problem (recall definition 1). The exact-solve error for MAML is not known in the literature; in particular, even as $\delta \to 0$ it is not evident if the approx-solve solution tends to the exact-solve solution, unless further regularity conditions are imposed. The approx-solve error for MAML is 0, ignoring finite-precision and numerical issues, since it backpropagates through the path. Truncated backprop [53] also invokes $\mathcal{A}lg_i$ to obtain a $\delta$-approximate solution but instead performs a truncated or partial back-propagation so that it uses a smaller number of iterations when computing the gradient through the path of $\mathcal{A}lg_i(\boldsymbol{\theta})$. Exact-solve error for truncated backprop is also not known, but a small approx-solve error can be obtained with less memory than full back-prop. We use Prop 3.1 of Shaban et al. [53] to provide a guarantee that leads to an $\epsilon$–accurate approximation of the full-backprop (i.e. MAML) gradient. It is not evident how accurate the truncated procedure is when an accelerated method is used instead. Finally, our iMAML algorithm also invokes an approximate solver $\mathcal{A}lg_i$ rather than $\mathcal{A}lg_i^\star$. However, importantly, we guarantee a small exact-solve error even though we do not require access to $\mathcal{A}lg_i^\star$. Furthermore, the iMAML algorithm also requires substantially less memory. Up to small constant factors, it only utilizes the memory required for computing a single gradient of $\hat{\mathcal{L}}_i(\cdot)$.

# E  Theoretical Results and Proofs

**Lemma 1, restated.** Consider $\mathcal{A}lg_i^\star(\boldsymbol{\theta})$ as defined in Eq. 4 for task $\mathcal{T}_i$. Let $\boldsymbol{\phi}_i = \mathcal{A}lg_i^\star(\boldsymbol{\theta})$ be the result of $\mathcal{A}lg_i^\star(\boldsymbol{\theta})$. If $\left(\boldsymbol{I} + \frac{1}{\lambda}\nabla_\phi^2\hat{\mathcal{L}}_i(\boldsymbol{\phi}_i)\right)$ is invertible, then the derivative Jacobian is

$$\frac{d\mathcal{A}lg_i^\star(\boldsymbol{\theta})}{d\boldsymbol{\theta}} = \left(\boldsymbol{I} + \frac{1}{\lambda}\,\nabla_\phi^2\hat{\mathcal{L}}_i(\boldsymbol{\phi}_i)\right)^{-1}.$$

*Proof.* We drop the task $i$ subscripts in the proof for convenience. Since $\boldsymbol{\phi} = \mathcal{A}lg^\star(\boldsymbol{\theta})$ is the minimizer of $G(\boldsymbol{\phi}', \boldsymbol{\theta})$ in Eq. 4, the stationary point conditions imply that

$$\nabla_{\phi'}G(\boldsymbol{\phi}', \boldsymbol{\theta})\,|_{\phi'=\phi} = 0 \implies \nabla\hat{\mathcal{L}}(\boldsymbol{\phi}) + \lambda(\boldsymbol{\phi} - \boldsymbol{\theta}) = 0 \implies \boldsymbol{\phi} = \boldsymbol{\theta} - \frac{1}{\lambda}\,\nabla\hat{\mathcal{L}}(\boldsymbol{\phi}),$$

which is an implicit equation that often arises in proximal point methods. When the derivative exists, we can differentiate the above equation to obtain:

$$\frac{d\boldsymbol{\phi}}{d\boldsymbol{\theta}} = \boldsymbol{I} - \frac{1}{\lambda}\,\nabla^2\hat{\mathcal{L}}(\boldsymbol{\phi})\frac{d\boldsymbol{\phi}}{d\boldsymbol{\theta}} \implies \left(\boldsymbol{I} + \frac{1}{\lambda}\,\nabla^2\hat{\mathcal{L}}(\boldsymbol{\phi})\right)\frac{d\boldsymbol{\phi}}{d\boldsymbol{\theta}} = \boldsymbol{I}.$$

which completes the proof. $\qquad\square$

Recall that:

$$G_i(\boldsymbol{\phi}', \boldsymbol{\theta}) := \hat{\mathcal{L}}_i(\boldsymbol{\phi}') + \frac{\lambda}{2}\,\|\boldsymbol{\phi}' - \boldsymbol{\theta}\|^2.$$

**Assumption 2.** *(Regularity conditions) Suppose the following holds for all tasks $i$:*

  1. $\mathcal{L}_i(\cdot)$ *is $B$ Lipshitz and $L$ smooth.*

2. *For all $\boldsymbol{\theta}$, $G_i(\cdot, \boldsymbol{\theta})$ is both a $\beta$-smooth function and a $\mu$-strongly convex function. Define:*

$$\kappa := \frac{\beta}{\mu}\,.$$

3. *$\hat{\mathcal{L}}_i(\cdot)$ is $\rho$-Lipshitz Hessian, i.e. $\nabla^2 \hat{\mathcal{L}}_i(\cdot)$ is $\rho$-Lipshitz.*

4. *For all $\boldsymbol{\theta}$, suppose the arg-minimizer of $G_i(\cdot, \boldsymbol{\theta})$ is unique and bounded in a ball of radius $D$, i.e. for all $\boldsymbol{\theta}$,*

$$\|\mathcal{A}lg_i^\star(\boldsymbol{\theta})\| \leq D\,.$$

**Lemma 3.** *(Implicit Gradient Accuracy) Suppose Assumption 2 holds. Fix a task $i$. Suppose that $\phi_i$ satisfies:*

$$\|\phi_i - \mathcal{A}lg_i^\star(\boldsymbol{\theta})\| \leq \delta$$

*and that $\boldsymbol{g}_i$ satisfies:*

$$\left\| \boldsymbol{g}_i - \left( \boldsymbol{I} + \frac{1}{\lambda}\nabla^2\hat{\mathcal{L}}_i(\phi) \right)^{-1} \nabla_\phi \mathcal{L}_i(\phi) \right\| \leq \delta'\,.$$

*Assuming that $\delta < \mu/(2\rho)$, we have that:*

$$\|\boldsymbol{g}_i - \boldsymbol{d_\theta}\mathcal{L}_i(\mathcal{A}lg_i^\star(\boldsymbol{\theta}))\| \leq \left( 2\frac{\lambda\rho}{\mu^2}B + \frac{\lambda L}{\mu} \right)\delta + \delta'$$

*Proof.* First, observe that:

$$\boldsymbol{d_\theta}\mathcal{L}_i(\mathcal{A}lg_i^\star(\boldsymbol{\theta})) = \left( \boldsymbol{I} + \frac{1}{\lambda}\nabla^2\hat{\mathcal{L}}_i(\mathcal{A}lg_i^\star(\boldsymbol{\theta})) \right)^{-1}\nabla_\phi\mathcal{L}_i(\mathcal{A}lg_i^\star(\boldsymbol{\theta}))$$

For notational convenience, we drop the $i$ subscripts within the proof. We have:

$$\|\boldsymbol{d_\theta}\mathcal{L}(\mathcal{A}lg^\star(\boldsymbol{\theta})) - \boldsymbol{g}\|$$
$$\leq \|\boldsymbol{d_\theta}\mathcal{L}(\mathcal{A}lg^\star(\boldsymbol{\theta})) - \left( \boldsymbol{I} + \frac{1}{\lambda}\nabla^2\hat{\mathcal{L}}(\phi) \right)^{-1}\nabla_\phi\mathcal{L}(\phi)\| + \delta'$$
$$\leq \|\boldsymbol{d_\theta}\mathcal{L}(\mathcal{A}lg^\star(\boldsymbol{\theta})) - \left( \boldsymbol{I} + \frac{1}{\lambda}\nabla^2\hat{\mathcal{L}}(\phi) \right)^{-1}\nabla_\phi\mathcal{L}(\mathcal{A}lg^\star(\boldsymbol{\theta}))\| + $$
$$\| \left( \boldsymbol{I} + \frac{1}{\lambda}\nabla^2\hat{\mathcal{L}}(\phi) \right)^{-1}(\nabla_\phi\mathcal{L}(\mathcal{A}lg^\star(\boldsymbol{\theta})) - \nabla_\phi\mathcal{L}(\phi))\| + \delta'$$

where the first inequality uses the triangle inequality.

We now bound each of these terms. For the second term,

$$\| \left( \boldsymbol{I} + \frac{1}{\lambda}\nabla^2\hat{\mathcal{L}}(\phi) \right)^{-1}(\nabla_\phi\mathcal{L}(\mathcal{A}lg^\star(\boldsymbol{\theta})) - \nabla_\phi\mathcal{L}(\phi))\|$$
$$\leq \| \left( \boldsymbol{I} + \frac{1}{\lambda}\nabla^2\hat{\mathcal{L}}(\phi) \right)^{-1}\|\|\nabla_\phi\mathcal{L}(\mathcal{A}lg^\star(\boldsymbol{\theta})) - \nabla_\phi\mathcal{L}(\phi)\|$$
$$\leq \lambda L\| \left( \lambda\boldsymbol{I} + \nabla^2\hat{\mathcal{L}}(\phi) \right)^{-1}\|\|\mathcal{A}lg^\star(\boldsymbol{\theta}) - \phi\|$$
$$= \lambda L\|\nabla_\phi^2 G(\phi, \boldsymbol{\theta})^{-1}\|\|\mathcal{A}lg^\star(\boldsymbol{\theta}) - \phi\|$$
$$\leq \frac{\lambda L}{\mu}\delta$$

where we the second inequality uses that $\nabla_\phi\mathcal{L}$ is $L$-smooth and the final inequality uses that $G$ is $\mu$ strongly convex.

For the first term, we have:

$$\|\boldsymbol{d_\theta}\mathcal{L}(Alg^\star(\boldsymbol{\theta})) - \left(\boldsymbol{I} + \frac{1}{\lambda}\nabla^2\hat{\mathcal{L}}(\boldsymbol{\phi})\right)^{-1}\nabla_{\boldsymbol{\phi}}\mathcal{L}(Alg^\star(\boldsymbol{\theta}))\|$$

$$= \|\left(\left(\boldsymbol{I} + \frac{1}{\lambda}\nabla^2\hat{\mathcal{L}}(Alg^\star(\boldsymbol{\theta}))\right)^{-1} - \left(\boldsymbol{I} + \frac{1}{\lambda}\nabla^2\hat{\mathcal{L}}(\boldsymbol{\phi})\right)^{-1}\right)\nabla_{\boldsymbol{\phi}}\mathcal{L}(Alg^\star(\boldsymbol{\theta}))\|$$

$$\leq \lambda\|\left(\lambda\boldsymbol{I} + \nabla^2\hat{\mathcal{L}}(Alg^\star(\boldsymbol{\theta}))\right)^{-1} - \left(\lambda\boldsymbol{I} + \nabla^2\hat{\mathcal{L}}(\boldsymbol{\phi})\right)^{-1}\|B,$$

using that $\nabla_{\boldsymbol{\phi}}\mathcal{L}$ is $B$ Lipshitz. Now let

$$\Delta := \nabla^2\hat{\mathcal{L}}(Alg^\star(\boldsymbol{\theta})) - \nabla^2\hat{\mathcal{L}}(\boldsymbol{\phi}), \quad M := \nabla^2_{\boldsymbol{\phi}}G(\boldsymbol{\phi},\boldsymbol{\theta}) = \lambda\boldsymbol{I} + \nabla^2\hat{\mathcal{L}}(\boldsymbol{\phi})$$

Due to that $\nabla^2\hat{\mathcal{L}}(\cdot)$ is Lipshitz Hessian, $\|\Delta\| \leq \rho\delta$. Also, by our assumption on $\delta$, we have that:

$$\|M^{-1}\Delta\| \leq \|\Delta\|/\mu \leq \rho\delta/\mu \leq 1/2,$$

which implies that $\|\left(\boldsymbol{I} + M^{-1}\Delta\right)^{-1}\| \leq 2$. Hence,

$$\|\left(\lambda\boldsymbol{I} + \nabla^2\hat{\mathcal{L}}(Alg^\star(\boldsymbol{\theta}))\right)^{-1} - \left(\lambda\boldsymbol{I} + \nabla^2\hat{\mathcal{L}}(\boldsymbol{\phi})\right)^{-1}\|$$

$$= \|\left(M + \Delta\right)^{-1} - M^{-1}\|$$

$$\leq \|M^{-1}\|\|\left(\boldsymbol{I} + M^{-1}\Delta\right)^{-1} - \boldsymbol{I}\|$$

$$= \|M^{-1}\|\|\left(\boldsymbol{I} + M^{-1}\Delta\right)^{-1}\left(\boldsymbol{I} - \left(\boldsymbol{I} + M^{-1}\Delta\right)\right)\|$$

$$\leq \|M^{-1}\|\|\left(\boldsymbol{I} + M^{-1}\Delta\right)^{-1}\|\|M^{-1}\Delta\|$$

$$\leq \frac{1}{\mu}\cdot 2 \cdot \frac{\rho\delta}{\mu} = 2\frac{\rho}{\mu^2}\delta.$$

The proof is completed by substitution. $\qquad\square$

**Theorem 2.** *(Approximate Implicit Gradient Computation) Suppose Assumption 2 holds. Fix a task i. Let*

$$B_1 \quad := \quad 2\frac{\lambda\rho}{\mu^2}B + \frac{\lambda L}{\mu}$$

*Suppose Nesterov's accelerated gradient descent algorithm is used to compute $\boldsymbol{\phi}$ (as desired in Algorithm 2), using a number of iterations that is:*

$$2\sqrt{\kappa}\,\log\left(8\kappa D\left(\frac{B_1}{\epsilon} + \frac{\rho}{\mu}\right)\right)$$

*and suppose Nesterov's accelerated gradient descent algorithm (or the conjugate gradient algorithm [2]) is used to compute $\boldsymbol{g}_i$ using a number of iterations that is:*

$$2\sqrt{\kappa}\,\log\left(4\kappa\frac{(\lambda/\mu)B}{\epsilon}\right).$$

*We have that:*

$$\|\boldsymbol{g}_i - \boldsymbol{d_\theta}\mathcal{L}_i(Alg_i^\star(\boldsymbol{\theta}))\| \leq \epsilon.$$

*Proof.* The result will follow from the guarantees in Lemma 2. Specifically, let us set $\delta = \min\{\epsilon/(2B_1), \mu/(2\rho)\}$ and $\delta' = \epsilon/2$. To ensure the bound of $\delta$, by Lemma 3, it suffices to use a number of iterations that is bounded by:

$$2\log\left(2\kappa\frac{\|D\|}{\delta}\right) \leq 2\sqrt{\kappa}\,\log\left(8\kappa D\left(\frac{B_1}{\epsilon} + \frac{\rho}{\mu}\right)\right)$$

To ensure the bound of $\delta'$, the algorithm will be solving the sub-problem in Equation 7. First observe that in the context of in Lemma 2, note that $\|x^\star\| = \| \left( \boldsymbol{I} + \frac{1}{\lambda} \nabla^2 \hat{\mathcal{L}}_i(\boldsymbol{\phi}) \right)^{-1} \nabla \mathcal{L}_i(\boldsymbol{\phi})\| \leq (\lambda/\mu)B$, and so it suffices to use a number of iterations that is bounded by:

$$2 \log \left( 2\kappa \frac{\|x^\star\|}{\delta} \right) \leq 2 \log \left( 4\kappa \frac{(\lambda/\mu)B}{\epsilon} \right),$$

which completes the proof. $\qquad\qquad\qquad\qquad\qquad\qquad\qquad\qquad\qquad\qquad\qquad\qquad\qquad\square$

Finally, we present a corollary of previous theorem that shows that iMAML finds approximate stationary points due to controllable error in gradient computation.

**Corollary 1.** *(iMAML finds stationary points) Suppose the conditions of Theorem 1 hold and that $F(\cdot)$ is an $L_F$ smooth function. Then the implicit MAML algorithm (Algorithm 1), when the batch size is $M$ (so that we are doing gradient descent), will find a point $\boldsymbol{\theta}$ such that:*

$$\|\nabla F(\boldsymbol{\theta})\| \leq \epsilon$$

*in a number of calls to* `Implicit-Meta-Gradient` *that is at most $\frac{4ML_f(F(0) - \min_{\boldsymbol{\theta}} F(\boldsymbol{\theta}))}{\epsilon^2}$. Furthermore, the total number of gradient computations (of $\nabla \hat{\mathcal{L}}_i$) is at most $\tilde{O}\left( M\sqrt{\kappa} \frac{L_f(F(0) - \min_{\boldsymbol{\theta}} F(\theta))}{\epsilon^2} \log \left( \frac{poly(\kappa, D, B, L, \rho, \mu, \lambda)}{\epsilon} \right) \right)$, and only $\tilde{O}(\text{Mem}(\nabla \hat{\mathcal{L}}_i))$ memory is required throughout.*

# F  Experiment Details

Here, we provide additional details of the experimental set-up for the experiments in Section 4. All training runs were conducted on a single NVIDIA (Titan Xp) GPU.

## F.1  Synthetic Experiments

For the synthetic experiments, we consider a linear regression problem. We consider parametric models of the form $h_{\boldsymbol{\phi}}(\mathbf{x}) = \boldsymbol{\phi}^T \mathbf{x}$, where $\mathbf{x}$ can either be the raw inputs or features (e.g. Fourier features) of the input. For task $\mathcal{T}_i$, we can equivalently write a quadratic objective that represents the task loss as:

$$\hat{\mathcal{L}}_i(\boldsymbol{\phi}) = \frac{1}{2}\mathbb{E}_{(\mathbf{x},\mathbf{y}) \sim \mathcal{D}_i^{\text{tr}}} \left[ \|h_{\boldsymbol{\phi}}(\mathbf{x}) - \mathbf{y}\|^2 \right] = \frac{1}{2}\boldsymbol{\phi}^T A_i \boldsymbol{\phi} + \boldsymbol{\phi}^T b_i,$$

where $A_i = \mathbb{E}_{(\mathbf{x},\mathbf{y}) \sim \mathcal{D}_i^{\text{tr}}} \left[ \mathbf{x}\mathbf{x}^T \right]$ and $b_i = \mathbb{E}_{(\mathbf{x},\mathbf{y}) \sim \mathcal{D}_i^{\text{tr}}} \left[ \mathbf{x}^T \mathbf{y} \right]$. Thus, the inner level objective and corresponding minimizer can be written as:

$$G_i(\boldsymbol{\phi}', \boldsymbol{\theta}) = \frac{1}{2}{\boldsymbol{\phi}'}^T A_i \boldsymbol{\phi}' + {\boldsymbol{\phi}'}^T b_i + \frac{\lambda}{2}(\boldsymbol{\phi}' - \boldsymbol{\theta})^T(\boldsymbol{\phi}' - \boldsymbol{\theta})$$

$$\mathcal{A}lg_i^\star(\boldsymbol{\theta}) = (A_i + \lambda \boldsymbol{I})^{-1} (\lambda \boldsymbol{\theta} - b_i)$$

Thus, the exact meta-gradient can be written as

$$\boldsymbol{d}_{\boldsymbol{\theta}} \mathcal{L}_i(\mathcal{A}lg_i^\star(\boldsymbol{\theta})) = \lambda(A_i + \lambda \boldsymbol{I})^{-1}\nabla_{\boldsymbol{\phi}} \mathcal{L}_i(\boldsymbol{\theta}) \mid_{\boldsymbol{\phi} = \mathcal{A}lg_i^\star(\boldsymbol{\theta})}.$$

We compare this gradient with the gradients computed by the iMAML and MAML algorithms. We considered the case of $\mathbf{x} \in \mathbb{R}^{50}$, $\mathbf{y} \in \mathbb{R}$, $\lambda = 5.0$, and $\kappa = 50$, for the presented results.

## F.2  Omniglot and Mini-ImageNet experiments

We follow the standard training and evaluation protocol as in prior works [50, 57, 15].

**Omniglot Experiments**  The GD version of iMAML uses 16 gradient steps for 5-way 1-shot and 5-way 5-shot settings, and 25 gradient steps for 20-way 1-shot and 20-way 5-shot settings. A regularization strength of $\lambda = 2.0$ was used for both. 5 steps of conjugate gradient was used to compute the meta-gradient for each task in the mini-batch, and the meta-gradients were averaged before taking a step with the default parameters of Adam in the outer loop.

The Hessian-free version of MAML proceeds by using Hessian-free or Newton-CG method for solving the inner optimization problem (with respect to $\phi$) with objective $G_i(\phi, \boldsymbol{\theta})$. This method proceeds by constructing a local quadratic approximation to the objective and approximately computing the Newton direction with conjugate gradient. 5 CG steps are used for this process in our experiments. This allows us to compute the search direction, following which a step size has to be picked. We pick the step size through line-search. This procedure of computing the approximate Newton direction and linesearch is repeated 3 times in our experiments to solve the inner optimization problem well.

**Mini-ImageNet** For the GD version of iMAML, 10 GD steps were used with regularization strength of $\lambda = 0.5$. Again, 5 CG steps are used to compute the meta-gradient. Similarly, in the Hessian-Free variant, we again use 5 CG steps to compute the search direction followed by line search. This process is repeated 3 times to solve the inner level optimization. Again, to compute the meta-gradient, 5 steps of CG are used.

## Footnotes

[2]The conjugate gradient descent algorithm also suffices and give a slightly improved iteration complexity in terms of log factors.