[Reviews · NeurIPS 2019]

Reviewer 1



Please find my concerns for this approach here: 1. There are too many strong continuity assumptions in the proof of Lemma 1. In reality, such assumptions may prove to be somewhat acceptable for simple datasets, but may not generalize to more complex scenarios. I would like to see a simple frequentist study showing how many times these assumptions are violated in reality. One such approach could be simply randomly evaluating the assumed continuities in a real-world setup. 2. Complementary to the above, the paper would be more impactful if authors study a real-world complicated task and show superior performance of their approach. Furthermore, study of failure cases would also be greatly appreciated. Table 2 shows an extremely saturated task, which only leaves the reviewer in doubt. Aside from a task already in 95+ performance, I would like to see the proposed model be challenged in a more complex setup. 3. What is the optimizer for MAML in Table 2? 4. Authors highlight that MAML may have gradient instabilities @274 which naturally the reader connects that to continuity of the gradients and the inner loop, but then wouldn't this also break your claims about continuity? What would happen if the assumptions are not true in your approach? Methods like MAML are desirable since they will work regardless of the exact complexity of the data and continuities across the optimization path. 5. Line 172 needs citation. It is not clear why line search methods would in fact fail.

Reviewer 2



The paper tackles the problem of efficiently differentiating through the inner-loop in meta-learning (mainly for few-shot classification/regression) so as the learn the optimal meta-learner, ie. the "optimal" initial model from which a new model can learned with few examples. To this end, the paper gives an excellent definition of one of the main problems in existing meta-learning algorithms, ie. the difficulty of computing gradients over the compute chain of the meta-learning algorithm's inner loop. On the positive side, the propose technique is simple, novel and clear; presented with neat arguments, and derived in a theoretically sound way. I really enjoyed reading the paper even if I am familiar with the problem being tackled. It is also good to see that the paper improves over directly relevant MAML-based baselines, even if it is not achieving state-of-the-art results. On the negative side, I am quite unimpressed with the experimental analysis & verification of the proposed method. The problems I see are as follows: - The paper makes one important speculative argument on the reasons for improving over MAML, even in the cases where MAML gives the exact inner-loop gradient (albeit being inefficient): the paper claims that MAML works worse due to numerical instability, but that's not verified! - The following question is not answered: would it still behave badly if you were to work implement MAML with infinite precision OR by scaling the data to minimize numerical problems? Or something else is going on, in terms of the improvements brought by using the proposed implicit gradient technique, when the exact same inner loop is being used as in MAML? - It would be very curious to see the accuracy of the implicit gradient by comparing that against exact (MAML) gradients. How does the implicit gradient's accuracy depend on the number of steps being computed? - I am surprised to see that Figure 2 presents iMAML with only 20 inner update steps. Why? Given that the performance of MAML baselines vary rather significantly depending on the number of steps, I find it truly essential to do the same kind of analysis with iMAML. - The choice of the datasets is just poor. Sigmoid-experiments are fine for understanding the algorithm. But Omniglot is pretty much saturate with accuracy values at around 98% in most settings. Why not some other commonly used benchmark, like mini-imagenet (with or without episodic training) or imagenet-1k, ideally in generalized few-shot learning setting? Incorporation of at least one "more modern" benchmark would significantly improve the experimental analysis. - Finally, although not very critical, an empirical comparison in terms of training speed and memory use between MAML and iMAML would be very nice to see for a few different inner loop optimizers.

Reviewer 3



The paper is well written and well structured. It clearly references existing work on (this type of) meta-learning, and gives an excellent overview of the area. The concept of "implicit differentiation" probably deserves a slightly better introduction, explanation + more references, given its relative importance to the main contribution. Apart from this slight exception, each section of the paper is well presented, concise, and easy to understand. I especially appreciated the references + background section, and the interpretation of iMAML relates to, builds on, differs from, and generalises previous work. I wish the paper were a bit clearer on what iMAML "gives up" compared to normal MAML, e.g. by pointing out situations where MAML might succeed and iMAML fila (and vice versa). I also would have wished for more detail + resolution in the experimental section, and perhaps for a slightly more thorough experimental comparison, or a stronger demonstration of superior performance (e.g. a wall-clock measurements of CPU- and memory usage). Nonetheless, it is an excellent paper that I enjoyed reading and would recommend to accept at NeurIPS 2019.

[Author Response · NeurIPS 2019]



Figure 1: Grad Error

Figure 2: Compute and memory time

Table 1 : MiniImagenet results.

| Algorithm | 5-way 1-shot |
|---|---|
| MAML | $48.70 \pm 1.84$ % |
| first-order MAML | $48.07 \pm 1.75$ % |
| iMAML GD (ours) | $48.96 \pm 1.84$ % |
| iMAML HF (ours) | $49.30 \pm 1.88$ % |

We thank the reviewers for the thoughtful feedback! The primary concern of R1 and R2 was a comparison on a more complex dataset. **We have added a comparison on MiniImagenet in Table 1,** and will include this and 5-shot results to the revised version. Again, we find that both the gradient descent (GD) and Hessian-free (HF) versions of the iMAML algorithm perform better than MAML.

**Reviewer #1:** Thank you for the thoughtful questions!

1. Lemma 1 only requires convexity of $G$ which is easily realizable since the regularization strength ($\lambda$) is under our control. Second order smoothness is needed only for finite-time analysis (Theorem 1), and this assumption is made in a number of optimization works works (e.g. Nesterov & Polyak '06, Jin et al. '17, '19, and references therein). We **do not** require convexity of $\mathcal{L}$ anywhere. We also emphasize that, to our knowledge, this is the first finite-time analysis of bilevel optimization problems, which we feel is an important fundamental contribution.
   Furthermore, regularity conditions are often needed for analysis but not to run the algorithm. Successful algorithms such as momentum and AdaGrad are well understood only for convex problems, but are a staple in modern deep learning. Similarly, iMAML shows promising empirical results.
2. We added results on the MiniImagenet domain above.
3. MAML uses GD in the inner level and Adam in the outer level. iMAML also uses Adam in the outer level.
4. Assumptions vs practice was addressed along with (1) above. Please see R2.1 and R2.2 for reasons as to why iMAML might perform better than MAML. On top of these, finite precision considerations might impact practical performance. While understanding the impact of finite precision is an interesting question, it is outside the scope of this work, and orthogonal to our main contributions. We will re-word line 274 appropriately in the revised version.
5. Line 172 refers to MAML, which backpropagates through the optimization path. There is no notion of derivatives through discontinuities or non-differentiable operations like linesearch. We will elaborate in the revised version.

**Reviewer #2:** Thank you for the valuable comments. We will strive to incorporate all of them for the revised version. Due to space constraints, we address a subset here.

1. Superior performance of iMAML is likely due to more GD steps or due to more powerful optimizer (HF). MAML cannot handle both of these due to memory constraints, requirements of higher-order derivatives, and inability to differentiate through linesearch. We will make this clear in the revision.
2. Even in cases where MAML can be implemented without the above limitations (e.g. small problems), iMAML can better approximate the exact bi-level gradient with finite iterations when the Hessian is smooth (Theorem 1 analyzes this explicitly). To illustrate this, Figure 1 presents a synthetic regression example using Fourier-features of inputs (thus predictions are non-linear in inputs, but linear in parameters). This allows us to analytically compute the exact meta-gradient and compare different algorithms with it. The condition number ($\kappa$) is large, thereby necessitating many GD steps. We find that both iMAML and MAML asymptotically match the exact meta-gradient, but iMAML computes a better approximation in finite iterations.
3. Figure 2 presents compute and memory trade-off for MAML and iMAML (on 20-way-5-shot Omniglot). Memory for iMAML is based on hessian-vector product, and is same for GD and HF, and indipendent of number of CG iterations. Memory for MAML grows linearly in grad steps, quickly hitting GPU capacity. Computational cost for iMAML is similar to FOMAML with a constant overhead for CG that depends on the number of CG steps. Compared to FOMAML, the compute cost of MAML grows at a faster rate. FOMAML requires only gradient computations, while backpropagating through GD (as done in MAML) requires a hessian-vector products at each iteration, which are more expensive than gradients.
4. On sinusoid, we verified that iMAML performs better across the number of GD steps. We will update the plot.

**Reviewer #3:** Thanks for the thoughtful comments. Memory and compute trade-offs are presented in Fig. 2 (please also see R2.3). We will include more details on implicit differentiation and trade-offs of different algorithms in the revised version. iMAML can support a wider set of optimizers, requires less memory than MAML, and has comparable compute requirements. Thus, iMAML is beneficial in cases requiring extended or complex optimization loops.

[Meta-Review · NeurIPS 2019]

This paper elaborates MAML with implicit gradients, avoiding long inner-loop chains appearing in the MAML. The idea of incorporating implicit differentiation into MAML is interesting, without any doubt. In general, the paper is well written, although it is not clear what iMAML should give up. During the discussion period, two of reviewers were positive and even one reviewer raised his/her score. However, one reviewer declined to strongly support this paper. While the paper is not perfect, I believe that the paper is deserved to be presented at NeurIPS.